# Modified Ultrasound-Guided Dorsal Quadratus Lumborum Block in Cat Cadavers

**DOI:** 10.3390/ani13243798

**Published:** 2023-12-09

**Authors:** Gonzalo Polo-Paredes, Francisco G. Laredo, Francisco Gil, Marta Soler, Amalia Agut, Eliseo Belda

**Affiliations:** 1Departamento de Medicina y Cirugía Animal, Facultad de Veterinaria, Universidad de Murcia, 30100 Murcia, Spain; gpolo@um.es (G.P.-P.); laredo@um.es (F.G.L.); mtasoler@um.es (M.S.); amalia@um.es (A.A.); 2Departamento de Anatomía y Anatomía Patológica Comparada, Facultad de Veterinaria, Universidad de Murcia, 30100 Murcia, Spain; cano@um.es

**Keywords:** abdominal analgesia, fascial block, feline, locoregional anesthesia, quadratus lumborum, ultrasound guided

## Abstract

**Simple Summary:**

The quadratus lumborum (QL) block is an ultrasound-guided locoregional anesthetic technique which aims to provide analgesia to the abdomen. Previous studies in dog cadavers have shown that a dorsal injection point, between the psoas minor muscle and the vertebral body of L1, is able to stain the truncus sympathicus and the T13–L3 rami ventrales, potentially providing analgesia to the abdominal wall and viscera. This dorsal approach could be feasible in cats and is probably able to offer similar results to those in canine cadavers. To assess this objective, 10 cat cadavers were used, and a mixture of methylene blue and iopromide (0.4 mL kg^−1^) was injected dorsal to the QL muscle in the abovementioned location. Computerized tomography and anatomical dissection were employed to evaluate the injectate spread. Our results showed the feasibility of this approach in cat cadavers as a consistent staining of the truncus sympathicus (T13–L3) and the rami ventrales of the spinal nerves (T13–L3) was observed. These results are compatible with the induction of somatic and visceral analgesia of the abdomen, although the cranial abdominal wall may not be covered.

**Abstract:**

The quadratus lumborum (QL) block is an ultrasound-guided locoregional anesthesia technique which aims to provide analgesia to the abdomen. The main objective of this study was to assess a modified ultrasound-guided dorsal QL block in cat cadavers. For this purpose, a volume of 0.4 mL kg^−1^ of a mixture of iopromide and methylene blue was administered between the psoas minor muscle and the vertebral body (VB) of the first lumbar vertebra, and its distribution was assessed in thirteen cat cadavers. We hypothesized that this injection point would be feasible, offering a more cranial distribution of the injectate and a more consistent staining of the truncus sympathicus. The study was divided into two phases. Phase 1 consisted of an anatomical study (three cadavers were dissected). Phase 2 consisted of the ultrasound-guided administration of the injectate and the assessment of its distribution by computed tomography and anatomical dissection. The results showed a consistent distribution of contrast media within five (4–8) VBs from T10 to L5. Methylene blue stained three (2–6) rami ventrales, affecting T11 (10%), T12 (20%), T13 (60%), L1 (85%), L2 (95%) and L3 (65%). The truncus sympathicus was dyed in all cadavers with a spread of five (3–7) VBs. Finally, the splanchnicus major nerve was stained in all cadavers (100%). These results suggest that this technique could provide analgesia to the abdominal viscera and the abdominal wall, probably with the exception of the cranial aspects of the abdominal wall.

## 1. Introduction

Over the last decades, the use of locoregional anesthesia techniques has increased greatly in veterinary medicine [1]. These techniques have been found useful to reduce patient discomfort, postoperative pain, opioid consumption and hyperalgesia in humans [2]. As a consequence, these strategies reduce the overall cost of the procedures and the hospitalization period and are useful as part of Enhanced Recovery After Surgery (ERAS) protocols in human and veterinary medicine [3,4]. In addition, ultrasound-guided techniques have improved the efficacy and safety of the nerve blocks compared to blind or nerve stimulation techniques [5]. Fascial plane blocks consist of the administration of local anesthetics within different interfascial planes to target nerves traveling inside them [6]. Ultrasound-guided interfascial plane blocks are a safe and useful alternative for providing analgesia to large body areas [7].

In veterinary medicine, the need to provide effective analgesia to the abdomen has led to the implementation of different locoregional anesthesia techniques. Among them, the transversus abdominis plane (TAP) block [8] and the rectus sheath (RS) block [9] have been reported to desensitize the abdominal wall. As the gold standard, epidural anesthesia induces both somatic and visceral analgesia of the abdomen but impairs motor activity in the pelvic limbs [10]. The interfascial plane between the quadratus lumborum (QL) and psoas minor (Pm) muscles has been proposed as an adequate injection site for providing analgesia to the abdomen in humans [11]. In veterinary anesthesia, the QL block was first described in dog cadavers by Garbin et al. [12]. In recent years, different injection points and approaches to performing this block have been proposed in canine cadavers, offering similar results [13,14,15,16]. The aim of the QL block is to reach the rami ventrales of the last thoracic, first lumbar nerves and the truncus sympathicus [12,17]. This technique would have the advantage, over the TAP and the RS blocks, of providing anesthesia to the abdominal wall and viscera without impairing the locomotor ability of the patient, as occurs after epidural anesthesia.

The traditional QL block carried out at the level of the second lumbar (L2) vertebra has also been evaluated in feline cadavers with comparable results to those obtained from canine cadavers [18,19]. It is important to consider that the small size and higher systemic toxicity of local anesthetics in cats [20] make this species a challenge when a fascial plane block is employed. Thus, the use of more adjusted doses of local anesthetics is always desirable in feline patients.

The aim of this study was to evaluate the usefulness of an ultrasound-guided dorsal QL block approach, administering a volume of 0.4 mL kg^−1^ of a mixture of dye and contrast media between the Pm and the VB of L1. We hypothesized that the administration of the injectate in this dorsal location could be feasible and offer similar results to those obtained in canine cadaver studies.

## 2. Materials and Methods

This research was approved by the Biosafety Committee in Experimentation (CBE 556/2023) of the University of Murcia. A total of 17 cat cadavers were used. The bodies were donated to the University of Murcia once the cats had died or were euthanized for reasons unrelated to the study and frozen promptly after death. Cadavers were thawed at room temperature (20–22 °C) 48 h before utilization.

Cadavers were excluded from the study when their body condition score (BCS) was inferior to 3/9 or superior to 7/9, based on visual inspection and palpation of bone structures after thawing [21], or there was evidence of trauma/anatomical alteration in the thoracolumbar area.

The study was divided into two phases.

### 2.1. Phase 1: Anatomical Study

Three feline cadavers were used to anatomically study the thoracolumbar region from the ninth thoracic (T9) to the seventh lumbar (L7) vertebrae. After clipping the ventral and lateral walls of the thorax and abdomen, the cadavers were placed in dorsal recumbency. A midline incision of the skin was made, and the rectus abdominis (RA), obliquus externus abdominis (OEA), obliquus internus abdominis (OIA) and transversus abdominis (TA) muscles were dissected. The rami ventrales of the tenth (T10), eleventh (T11) and twelfth (T12) thoracic nerves were identified in the intercostal space and ventral to the costal arch and followed along the intercostal space. Likewise, the rami ventrales of the thirteenth (T13) thoracic nerve and the first (L1), second (L2) and third (L3) lumbar nerves were identified within the interfascial plane between the TA and the aponeurosis of the OIA. Then, an incision of the linea alba was performed, and the cadavers were eviscerated. The path of the previously described rami ventrales of the nerves was followed to its origin in the intervertebral foramen. The correlation between these nerves and the QL, Pm and psoas major (PM) muscles was assessed. Then, the splanchnicus major nerve and the truncus sympathicus were identified, and their correlation with the anatomic structures nearby was evaluated. The dissection of the three cadavers was performed by the same researchers (Francisco Gil and Gonzalo Polo-Paredes).

### 2.2. Phase 2

#### 2.2.1. Ultrasound-Guided Technique

In this phase, 14 cat cadavers were used. Two syringes with a total volume of 0.4 mL kg^−1^ each of a 1:1 solution of methylene blue (5 mg mL^−1^, Pancreac Quimica, AppliChem, Castellar del Vallès, Spain) and iopromide (300 mg mL^−1^, UltraVist300, Bayer, Berlin, Germany) were prepared.

The cadavers were clipped as previously described and positioned in lateral recumbency. A linear ultrasound probe of 3–13 Hz (MyLab Gamma, Esaote, Florence, Italy) was placed caudal to the last rib in a transversal view to the spine, and the epaxial and hypaxial musculature was identified. Then, the transducer was rotated cranially 30° in relation to the transverse axis of the spine. The final position of the probe was parallel to the caudal aspect of the rib (Figure 1). Firstly, the VB, cranial articular and transverse processes of L1 were identified. The probe was then glided slightly caudally until the transverse process of L1 disappeared. In this oblique position of the transducer, the spinous process and VB of L1, as well as the QL and Pm, were identified (Figure 2). Finally, a sonovisible needle (Ultraplex 10 mm, 30°, BBraun, Melsungen, Germany) was advanced “in plane” in a ventro-caudal to dorso-cranial direction until the tip of the needle contacted the VB of L1, and the injectate (0.4 mL kg^−1^) was administered (Figure 3). The same procedure was repeated in the contralateral hemiabdomen. All the injections were performed by the same researcher (Gonzalo Polo-Paredes).

#### 2.2.2. Computed Tomography (CT) Study

Immediately after the administration of the injectate in both hemiabdomens, computed tomography (CT) scans (dual-slice CT scanner, General Electric HiSpeed, General Electric Healthcare, Madrid, Spain) of the region between the eighth thoracic (T8) and the first sacral (S1) vertebra were carried out. Images were acquired with the cadavers placed in dorsal recumbency with their thoracic and pelvic limbs extended. Slice thickness was set to 3 mm, collimator pitch 1 and reconstructions interval with 50% overlap; mA was 100, and kVp was 120; standard soft tissues and bone reconstruction algorithms were used. Two certified radiologists (Marta Soler and Amalia Agut) evaluated the reformatted images and assessed the distribution and location of the contrast medium.

#### 2.2.3. Spread Study

Once the CT images were obtained, all the cadavers were taken immediately for anatomical dissection. A ventral midline incision was made, the linea alba was opened and a unilateral radical costotomy from the first to the twelfth rib was made. The thorax and abdomen viscera were carefully examined and both cavities eviscerated. The hypaxial muscles were exposed and the rami ventrales followed to their origins in the intervertebral foramen. Nerves were considered positively dyed when at least 1 cm of their length was stained in all their circumference. Finally, the presence and length of the staining of the splanchnicus major nerve and truncus sympathicus were evaluated. All the dissections were made by the same researchers (Francisco Gil and Gonzalo Polo-Paredes).

Statistical Analysis

The statistical descriptive test was performed using Microsoft Excel 365 software (Microsoft Corporation, Redmond, WA, USA) and Real Statistics Resource Pack software (release 7.6, Copyright 2013–2021, Charles Zaiontz, www.real-statistics.com). Normality was assessed using a Shapiro–Wilk test. The results are expressed as mean ± standard deviation (SD) (weight) or median (range) (BCS, spread of contrast media/colorant), as is appropriate (normal vs. non-normal distribution, respectively).

## 3. Results

### 3.1. Phase 1: Anatomical Study

Three cat cadavers (two European shorthairs and one Persian) were included in this phase of the study. They weighed 3.7, 3.75 and 1.57 kg. The BCS was 4/9 in all cats. A unilateral hematoma was observed in the hypaxial region in one cadaver during dissection, and only five out of six hemiabdomens were finally studied.

The QL was identified ventral to the transverse processes arising from the last three thoracic vertebrae. The Pm, with origin in the caudal border of T12, T13, L1, L2 and L3 VBs, laid contacting ventrally and medially with the QL and gathered caudally in a thin tendon. The PM arises from L2 and widens caudally in all animals (Figure 4).

The rami ventrales of T10–12 were identified emerging from the intervertebral foramina. They left the hypaxial region caudal to their respective ribs at the end of the first third of the intercostal space, running between the intercostales interni and externi muscles. Then, they ran cranioventrally, joining the caudal aspect of the rib approximately at 1 cm ventral to their origin.

The rami ventrales of T13–L3 were identified between the TA and OIA. The relationship between the nerves and the hypaxial muscles showed great variability. T13 ran dorsally to the QL or in the interfascial plane between the QL and Pm. The rami ventrales of L1, L2 and L3 were found between the bundles of QL, the interfascial plane of the QL and PM and the bundles of the PM. All the rami ventrales of L4 were observed between the bundles of the PM (Table 1). The truncus sympathicus and splanchnicus major nerve were identified in all the cadavers. The truncus sympathicus was observed bilaterally lying ventral to the VBs throughout the spine. The splanchnicus major nerve was located ventral to the VB of L1, caudal to the pillars of the diaphragm, and followed to its origin between the fifth (T5) and T9 thoracic VBs.

### 3.2. Phase 2

#### 3.2.1. Demographic Distribution

A total of 14 cat cadavers were enrolled in this phase, although four were excluded from the study (diaphragmatic rupture, high-volume peritoneal effusion, cachexia and a massive mammary mass). Ten cat cadavers (20 hemiabdomens) (seven females and three males) were finally used for this phase of the study. They weighed 2.99 ± 1.03 kg, with a BSC of 4 (3–5) out of 9 (Table 2).

#### 3.2.2. Ultrasound-Guided Technique

All the anatomical target structures were identified, and the needle visualized through its entire course in all the injections performed (20/20) in order to perform the injections. In 1 out of 20 hemiabdomens, the needle tip was advanced into the retroperitoneal cavity, but it could be redirected correctly afterwards.

#### 3.2.3. Computed Tomography Study

The contrast media were spread within five (4–8) VBs from T10 to L5 (Figure 5). At the injection point, iopromide was observed in the Pm muscle in all hemiabdomens (20/20). The spread of contrast media was more consistent at the level of T13 (90%), L1 (100%), L2 (100%) and L3 (95%), surrounding all the aspects of the QL and Pm. In the area between T10 and T13, iopromide was located ventromedial to the QL, and a more erratic distribution was observed in L4 and L5 segments. Contrast media were found in the TA plane in 18/20 hemiabdomens. Finally, a small amount of contrast was identified in the vertebral canal in three cadavers and in the retroperitoneal cavity (dorsal to the right kidney) in one cadaver.

#### 3.2.4. Spread Study

Methylene blue was observed within the hypaxial musculature with a cranio-caudal distribution, and the Pm was intramuscularly stained in all the hemiabdomens (20/20). The thoracolumbar aorta and the retroperitoneal cavity were also found to be dyed in 4 out of 20 and 1 out of 20 hemiabdomens, respectively. Colorant was not observed inside the peritoneal cavity in any of the cadavers. Methylene blue stained three (2–6) rami ventrales, consistently affecting T13 (60%), L1 (85%), L2 (95%) and L3 (65%). The rami ventrales of T11 (10%) and T12 (20%) were only dyed occasionally (Figure 6).

The truncus sympathicus was visualized in 19/20 hemiabdomens, with a colorant spread of five (3–7) VBs (non-normal distribution). Dyeing was consistently found at the level of the VBs of T12 (80%), T13 (95%), L1 (100%), L2 (100%), L3 (80%) and L4 (50%). Contrarily, VBs of T9 (5%), T10 (10%) and L5 (5%) were seldom dyed. All major splanchnicus nerves (20/20) were found to be stained (Figure 7).

## 4. Discussion

The results of this study showed that a dorsal approach to the QL block allows the administration of the injectate between the Pm and the VB of L1, and consistently stained the rami ventrales from T13 to L3, the truncus sympathicus and the splanchnicus major nerve in cat cadavers. Furthermore, this modified approach allows the visualization of the whole length of the needle and the injection point. According to these results, it could be inferred that this approach is safe and may provide effective analgesia to the abdomen, except for the cranial aspects of the abdominal wall.

Our anatomical observations are comparable to those previously described in the literature [22], evidencing the high prevalence of the interbundle nerve path of the rami ventrales of the nervi spinales within the thoracolumbar region. The variability observed regarding the path of these nerves led us to consider that the administration of the injectate closer to the VB of L1 might be clinically more effective. However, the anechoic acoustic shadowing of the transverse process of L1 can impede the visualization of its VB. For this reason, it was decided to modify the final position of the transducer compared with previous descriptions [18,19]. The oblique orientation of the array employed in our study allowed the visualization of the VB and the entire needle path as the “obstacle” produced by this acoustic shadow was avoided. This made it possible to administer the injectate in a more cranial position (L1) when compared with previous studies carried out in cats [18,19]. The injection point chosen in this study was employed by Marchina-Gonçalves et al. [14] in dog cadavers, and they observed consistent staining of the rami ventrales (L1–L3) as well as the truncus sympathicus.

The first reference to the QL block in cats was made by Argus et al. (2020) [23], who described in a case report the administration of ropivacaine in the interfascial plane between the QL and Pm. Recently, two cat cadaveric studies have been published regarding the QL block [18,19]. These authors reported that the administration of the injectate at the level of L2, between the QL and the Pm, was adequate to stain the rami ventrales between L1 and L3, the truncus sympathicus and the splanchnicus nerves [18]. Similarly, our study shows that a dorsal injection site, between the Pm and VB of L1, is a feasible approach to performing this block in cat cadavers, offering similar results to the interfascial QL block [17,18]. Our findings are also comparable with previous studies performed in dog cadavers where a similar dorsal approach to performing the QL block was employed [14,15]. In contrast, a recent study performed in rabbit cadavers showed a more caudal spread, staining up to the L6 rami ventrales, when a similar volume of injectate was administered [24].

A volume of 0.4 mL kg^−1^ of injectate per hemiabdomen was chosen to neither excessively increase the dose nor dilute a local anesthetic in a real scenario. The administration in live cats of a total volume of 0.8 mL kg^−1^, as employed in our study, would lead to a total dose of 2 mg kg^−1^ of bupivacaine or ropivacaine 0.25%. This final dose of local anesthetic was also chosen taking into account the desire not to exceed the recommended toxic dose of these local anesthetics in cats [20]. In addition, the use of this volume for the injections allowed a direct comparison of our results with those previously reported by Dos-Santos et al. [18,19]. It should also be considered that lower volumes may limit the injectate distribution and the length of the staining of the nerves. On the other hand, higher volumes could be safely used by diluting the local anesthetic. However, reducing local anesthetic concentration could decrease its effectivity in an in vivo setting [25].

The innervation of the abdominal wall is supplied by the rami ventrales of the nervi spinales from T10 to L3 [22]. Thus, to achieve a complete abdominal wall desensitization, dermatomes from T10 to L3 should be blocked. Our study showed a consistent staining of rami ventrales of these nerves from L1 to L3. Furthermore, visceral pain depends on the sensory afferent neurons, which travel along the same path as the sympathetic fibers. They diverge only when splitting into dorsal and ventral spinal roots [7]. Thus, staining of the truncus sympathicus and splanchnicus nerves would lead to a block of afferent visceral pain axons. Our results suggest that the technique described here to perform the QL block in cats could provide analgesia to the medium and caudal regions of the abdominal wall and to the main abdominal viscera. However, it would seem unreliable to provide analgesia to the more cranial aspects of the abdomen.

Several locoregional anesthetic techniques have been proposed to provide anesthesia to the abdomen. Among them, epidural anesthesia has been widely used to prevent and treat perioperative pain in abdominal surgeries in veterinary medicine [26,27,28]. One of the main limitations of this technique is the postoperative motor impairment of the pelvic limbs. In addition, other adverse effects such as emesis, pruritus, urinary retention and respiratory depression have been associated with it [27]. Using a QL block, all these side effects could be avoided. Comparing different fascial blocks, the TAP block has been shown to provide abdominal analgesia in cats undergoing elective ovariohysterectomy [8]. In contrast, RS block seemed to reduce isoflurane requirements during ovariectomy but not the scores of postoperative pain compared to the control group [9]. Theoretically, these two techniques can only provide analgesia to the abdominal wall and not to the viscera, although some human studies document that the TAP block could provide visceral analgesia [29,30]. The main advantage of the QL block over the TAP and RS blocks is the distribution of the local anesthetic over the truncus sympathicus, providing visceral analgesia to the abdomen. Recently, a clinical case report in cats undergoing cystotomy [23], a clinical study about this block in dogs undergoing ovariohysterectomy [31] and a case report of a minipig undergoing ovariectomy [32] have shown positive results. However, clinical trials in cats are still lacking.

Distribution of injectate in the transversus abdominis plane (TAP) was observed in some cadavers, as well as in the vertebral canal, in agreement with previous studies carried out in dog cadavers [15,16]. The observation of dye inside the TAP was not feasible during dissection due to the small size of these muscular layers in cat cadavers, but it was observed by CT imaging. Similarly, the observation of contrast inside the vertebral canal was assessed by CT imaging as hemilaminectomies were not conducted in the cadavers. Intramuscular injection into the Pm was also observed in all cases, probably due to the size of the bevel of the needles compared with the muscle. These facts could contribute to increasing the analgesic success ratio in an in vivo model as the spread of the injectate correlates with the path of the nerves. Compared to similar approaches described in dog cadavers [14,15], we did not observe staining of the rami ventrales of L4. On the contrary, our study showed a more cranial spread of the injectate, which reached the truncus sympathicus at the level of T9 and stained the T11 intercostal nerve occasionally. These findings could be explained by anatomical differences between species [22,33] and changes in the needle orientation and trajectory, which was dorso-cranial in the current study. Another observation to consider was the wider spread of colorant observed within the truncus sympathicus during dissection when compared with the CT images. These findings could be the result of the lapse of time that occurs between the CT and the dissection studies.

Our study has several limitations. The small number of cadavers could have biased the results. The distribution of the injectate can vary in an in vivo setting as a consequence of differences in hydration, muscle tension forces and the abdominal and thoracic wall respiratory movements. It should also be considered that the freezing and thawing of the cadavers could alter the spread of the injectate even more. In addition, the difference in the physical and chemical characteristics between the mixture of methylene blue/iopromide and the local anesthetics could also affect the spread of the injectate in live cats. The researcher who made the ultrasound-guided injections also performed the anatomical dissection studies. This aspect could have biased the dissection results. Finally, no large cat cadavers or cats with a BCS under 3/9 or over 7/9 were included here. This fact should be considered in future studies as differences in the size of the anatomical structures compared to the dimensions of the bevel of the needle, and the presence or absence of body fat deposits, may favor different spreading patterns of the injectate.

## 5. Conclusions

Our modified dorsal approach to the QL block, in which the injectate was administered between the Pm and the VB of L1, is feasible and can provide analgesia to the middle and caudal abdomen walls and to the abdominal viscera. A spread of injectate cranial to T13 was not consistently observed, so analgesia of the cranial aspects of the abdominal wall could not be reliably provided. Further studies are needed to assess the analgesic extension of this modified anesthetic technique in a real clinical setting in live cats.

## Figures and Tables

**Figure 1 animals-13-03798-f001:**
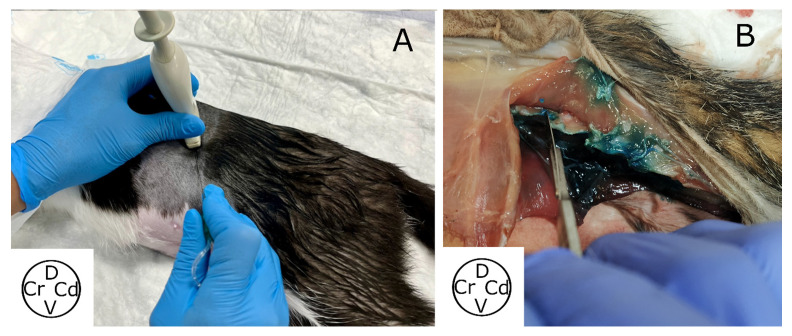
(**A**) Ultrasound-guided dorsal approach to the QL block in cat cadavers. Cat cadavers were positioned in lateral recumbency and the transducer positioned transversal to the spine at the L1 level and then rotated 30° cranially. The needle was advanced “in-plane”. (**B**) Anatomic reconstruction of the ultrasound beam used to perform the approach, avoiding bone structures. QL, quadratus lumborum; Cr, cranial; Cd, caudal; D, dorsal; V, ventral.

**Figure 2 animals-13-03798-f002:**
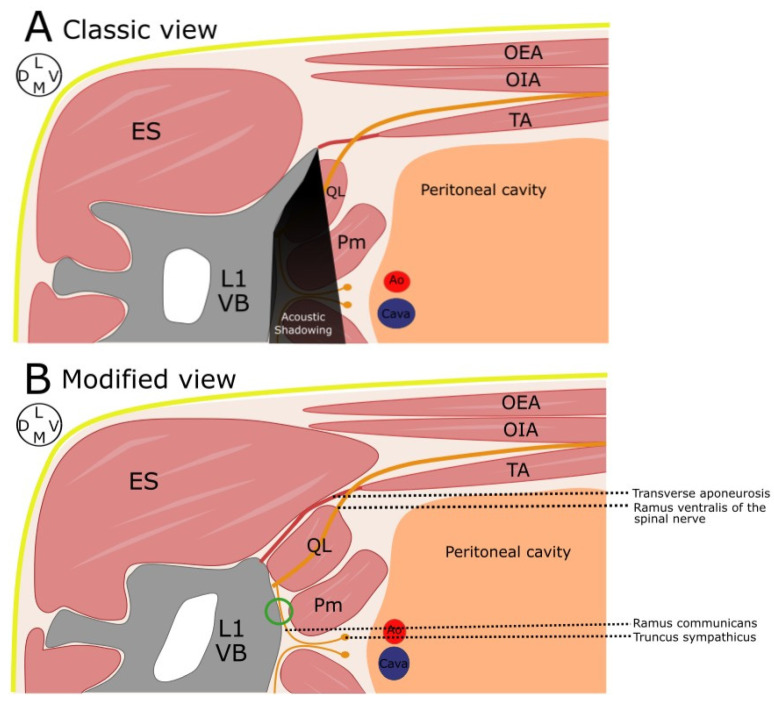
Schematic illustration of the L1 region in a cat placed in left lateral recumbency. (**A**) Classic view of the QL approach. (**B**) Modified view of the dorsal QL approach. The point of injection is represented by the green circle. Ao, aorta; ES, erector spinae muscles; OEA, obliquus externus abdominis muscle, OIA: obliquus internus abdominis muscle; TA, transversus abdominis muscle; Pm, psoas minor muscle; QL, quadratus lumborum muscle; L1 VB, vertebral body of L1; D, dorsal; V, ventral; L, lateral; M, medial.

**Figure 3 animals-13-03798-f003:**
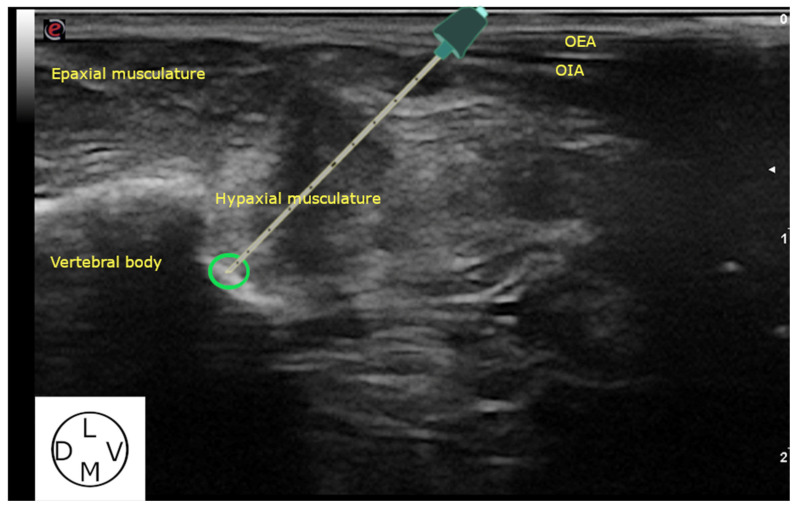
Ultrasound images of the modified approach to the QL block in cat cadavers. The needle should contact the vertebral body (green circle), characterized by a hyperechoic line creating an acoustic shadowing. OEA, obliquus externus abdominis muscle; OIA, obliquus internus abdominis muscle; D, dorsal; V, ventral L, lateral; M, medial.

**Figure 4 animals-13-03798-f004:**
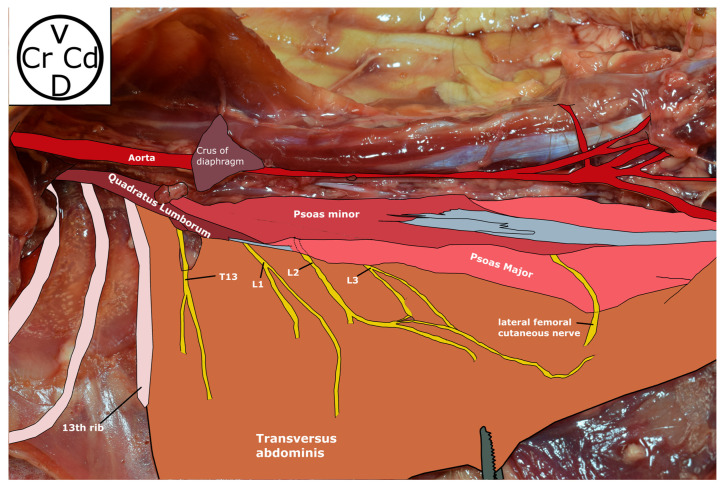
Schematic anatomical view of the region between T11 and L7 vertebrae. T13, L1, L2, L3 rami ventrales of T13, L1, L2 and L3 nerves, respectively. Cd, caudal; Cr, cranial; D, dorsal; V, ventral.

**Figure 5 animals-13-03798-f005:**
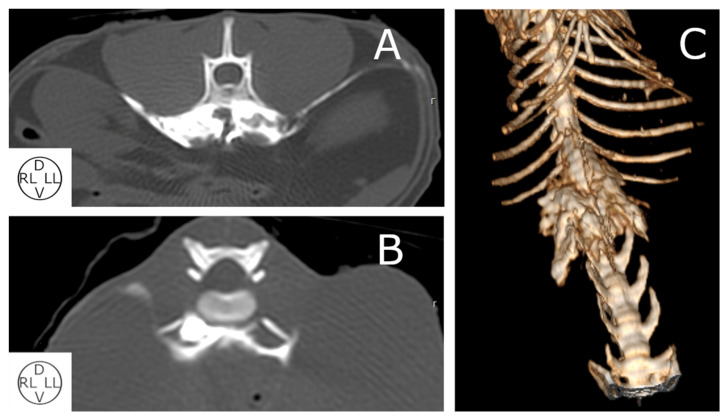
Computed tomographic images in transverse plane at the level of L2 with bone window (**A**,**B**) and 3D-VR reconstruction (**C**) of different cat cadavers after the administration of 0.4 mL kg^−1^ of a mixture of iopromide and methylene blue at the vertebral body of L1. (**A**) Contrast can be observed surrounding muscles and within the transversus abdominis plane. (**B**) Intramuscular spread of iopromide into Pm in the right hemiabdomen is visualized. (**C**) Volume-rendered three-dimensional reconstruction image of the thoracolumbar area between T12 and L4 vertebrae showing the distribution of the contrast. Pm, psoas minor muscle; D, dorsal; LL, left lateral; RL, right lateral; V: ventral.

**Figure 6 animals-13-03798-f006:**
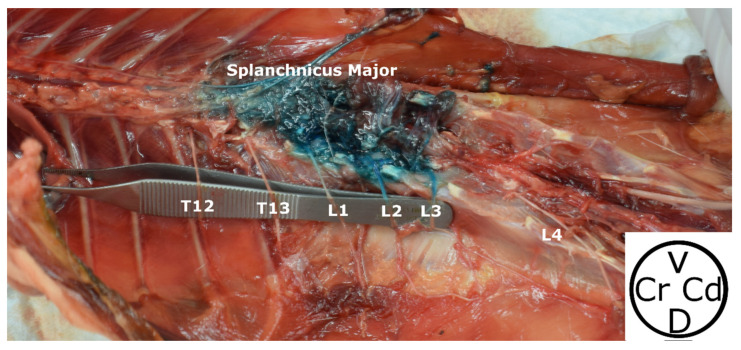
Anatomical dissection of the thoracolumbar area stained after methylene blue administration. T13, L1, L2, L3, L4 rami ventrales of T13, L1, L2, L3 and L4 nerves, respectively. Cr, cranial; Cd, caudal; D, dorsal; V, ventral.

**Figure 7 animals-13-03798-f007:**
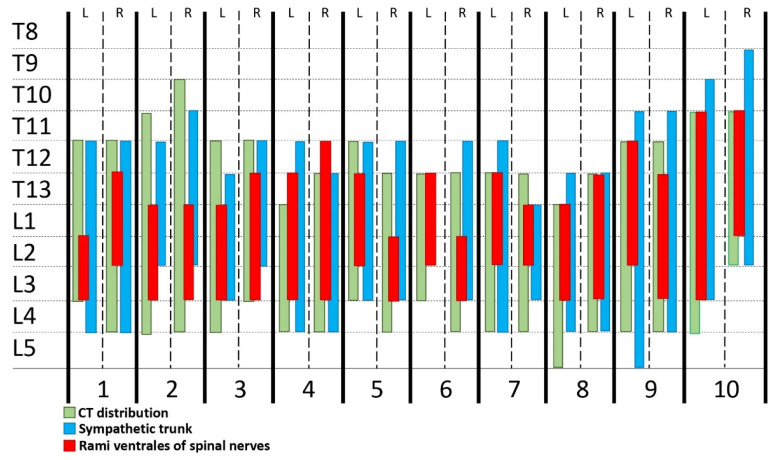
Staining of the rami ventrales of the spinal nerves and the truncus sympathicus evaluated by computed tomographic and anatomical dissection after the administration of 0.4 mL kg−1 of a mixture of methylene blue and iopromide by a modified dorsal QL approach. QL, quadratus lumborum block; L, left hemiabdomen; R, right hemiabdomen.

**Table 1 animals-13-03798-t001:** Description of the path of the rami ventrales of the spinal nerves through the hypaxial musculature. PM, psoas major muscle; Pm, psoas minor muscle; QL, quadratus lumborum muscle; T13, L1, L2, L3, L4 rami ventrales of T13, L1, L2, L3 and L4 nerves, respectively.

Muscles	Dorsal to QL	Through QL	Between QL and Pm	Through PM	Between QL and PM
Rami Ventrales					
T13	3/5		2/5		
L1		5/5			
L2		4/5			1/5
L3		3/5		1/5	1/5
L4				5/5	

**Table 2 animals-13-03798-t002:** Demographic distribution of the cat cadavers. BCS, body condition score.

Breed	Weight (kg)	BCS (1–9/9)
European Shorthair	3.37	4/9
European Shorthair	3.45	5/9
Siamese	2.20	3/9
European Shorthair	1.56	3/9
European Shorthair	2.17	4/9
European Shorthair	3.07	5/9
European Shorthair	5.38	5/9
European Shorthair	3.26	4/9
Persian	2.76	5/9
European Shorthair	2.70	4/9

## Data Availability

Data supporting the reported results can be sent to anyone interested by contacting the corresponding author.

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
