# Peer review of "Modified Ultrasound-Guided Dorsal Quadratus Lumborum Block in Cat Cadavers"

_animals, 2023, doi:10.3390/ani13243798_

Round 1

Reviewer 1 Report

Comments and Suggestions for Authors

Thank you for submitting this interesting paper. Despite there is literature describing this block in cats, the authors describe a different approach to have better visibility in the US image, similar to described previously in dogs. This type of studies are very important for clinical anaesthetists to improve the way they perform this block, to optimise its results at providing analgesia. I only ask major revision as I would like to read it back after changes have been made.

Line 45: replace “postoperatory” for “postoperative”

Line 46: replace "among them" for "in addition"

Line 47: replace "improve" for "have improved".

Line 53: replace "to resolve them" for "as a treatment".

Line 57: in humans?

Line 173: staining rather than dyeing might read better here, although dyeing is correct.

Line 181: the results were.

Lines 192-193: please use past tense throughout the sentence please -Laid from lays not lies, which I believe is to tell something that is not true.

Line 238: replace by “in all the injections performed”. I believe 20/20 were injections performed following the same approach?

Line 240: afterwards

Line 290: showed, use the past tense.

Line 291: stained

Line 293: according to these results

Line 306: delete “fact “and “it”

Line 345: spread of contrast in the vertebral canal: you cited a reference (14) . As far as I could see on that paper, states that there was no contrast on the epidural space? It would be good to discuss hypothesis of the injectate on this location.

Figure 4: A. iliaca ext: I think this is Spanish, please change to English.

Reviewer 2 Report

Comments and Suggestions for Authors

Dear authors,

Thank you very much for the preparation of your manuscript.about Quadratus lumborum Block in cat cadavers.

This study is an original study and provide basic information about anatomy and spread of this technique, which is useful for clinical implications.

Some minor comments:

Line 44-46 – please include that these studies are from humans. However, consider adding Campoy 2022 et al. about enhanced recovery after surgery in veterinary patients.

Line 52-53 can be deleted (including Reference Number 7) as not part of your topic and not necessary for the introduction. Rather comment on relevant locoregional anaesthesia techniques for abdominal procedures and point out the advantage of QL block.

Line 59-70: these are very detailed informations about the block technique and the reader is overwhelmed by the different strategies. I would suggest moving these parts to the discussion, but rather giving just a short introduction to the QL block itself and where they differ (as an overview rather than a discussion). Also missing specific advantage of QL block over other abdominal blocks.

Figure 1/Figure 2:

The figures are good, however, it is unclear for the reader what the classic and what the modified view of the QL block is. Why is figure 2A necessary? Wouldn’t it be good to introduce why to modify the classic view (and than maybe to stick to the modified view, as the classic view is not performed at all in this study)

2.2.2: a few more details for the CT descriptions are missing

Paragraph line 177 and following: choose of median or mean as more convenient? Please clarify? Statistics unclear whether any comparison was made? Or just descriptives?

 (eg CT vs dissection). Please rephrase and specify

Discussion:

Please include advantage of this technique over already published approaches?

When discussing volume, maybe comment on risk with higher or lower volume (not dose)

Also compare technique to the one described in rabbits (Torres Canto 2023).

Please include a short paragraph about analgesic efficiency of local blocks in cats as well (these are cadaveric studies, which are baseline), but it might be good to compare it to another block, where the established technique was actually used (even if not QL block)

Reviewer 3 Report

Comments and Suggestions for Authors

Congratulations to the writers. Really well described and written.

I have only one question:

- Regarding the Body Condition Score you use to exclude some cadavers, how do you perform this exam in a cadaver? After freezing/thawing, the tissues can be affected and this could affect too the BCS. It would be really useful to include the template for BCS or at least explain it. Actually, some of the limitations you describe are based on body condition. Could this affect the results?

Round 2

Reviewer 1 Report

Comments and Suggestions for Authors

Thank you for making these amendments. Congratulations